# Linear TreeShap

**Peng Yu**[1,2,5]    **Chao Xu**[1*]    **Albert Bifet**[2,4]    **Jesse Read**[3]

[1] University of Electronic Science and Technology of China
[2] LTCI, Télécom Paris, IP Paris    [3] LIX, Ecole Polytechnique, IP Paris
[4] AI Institute, University of Waikato    [5] Shopify
{peng.yu,albert.bifet}@telecom-paris.fr
cxu@uestc.edu.cn
jesse.read@polytechnique.edu

## Abstract

Decision trees are well-known due to their ease of interpretability. To improve accuracy, we need to grow deep trees or ensembles of trees. These are hard to interpret, offsetting their original benefits. Shapley values have recently become a popular way to explain the predictions of tree-based machine learning models. It provides a linear weighting to features independent of the tree structure. The rise in popularity is mainly due to TreeShap, which solves a general exponential complexity problem in polynomial time. Following extensive adoption in the industry, more efficient algorithms are required. This paper presents a more efficient and straightforward algorithm: Linear TreeShap. Like TreeShap, Linear TreeShap is exact and requires the same amount of memory.

## 1   Introduction

Machine learning in the industry has played more and more critical roles. For both business and fairness purposes, the need for explainability has been increasing dramatically. As one of the most popular machine learning models, the tree-based model attracted much attention. Several methods were developed to improve the interpretability of complex tree models, such as sampling-based local explanation model LIME[9], game-theoretical based Shapley value[10], etc. Shapley value gained particular interest due to both local and globally consistent and efficient implementation: TreeShap[5]. With the broad adoption of Shapley value, the industry has been seeking a much more efficient implementation. Various methods like GPUTreeShap[6] and FastTreeShap[12] were proposed to speed up TreeShap. GPUTreeShap primarily focuses on utilizing GPU to perform efficient parallelization. And FastTreeShap improves the efficiency of TreeShap by utilizing caching. All of them are empirical approaches lacking a mathematical foundation and are thus making them hard to understand.

We solve the exact Shapley value computing problem based on polynomial arithmetic. By utilizing the properties of polynomials, our proposed algorithm Linear TreeShap can compute the exact Shapley value in linear time. And there is no compromise in memory utilization.

### 1.1   Contrast with previous result

We compare the running time of our algorithm with previous results for a single tree in Table 1, since all current algorithms for ensemble of trees are applying the same algorithm to each tree individually.

---

*Corresponding author

36th Conference on Neural Information Processing Systems (NeurIPS 2022).

Let $S$ be the number of samples to be explained, $N$ the number of features, $L$ the number of leaves in the tree, and $D$ is the maximum depth of the tree. For simplicity, we assume every feature is used in the tree, and therefore $N = O(L)$. Also, $D \leq L$.

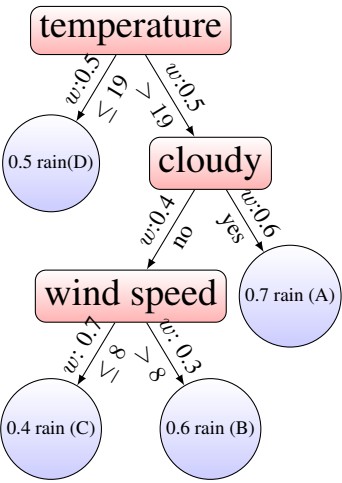

| Algorithm | Time Complexity | Space Complexity |
|:---:|:---:|:---:|
| Original TreeSHAP [5] | $O(SLD^2)$ | $O(D^2 + N)$ |
| Fast TreeSHAP v1 [12] | $O(SLD^2)$ | $O(D^2 + N)$ |
| Fast TreeSHAP v2 [12] | $O(L2^D D + SLD)$ | $O(L2^D)$ |
| Linear TreeSHAP | $O(SLD)$ | $O(D^2 + N)$ |

**Table 1:** Comparison of both computational and space complexity

**Figure 1:** An example decision tree $T_f$ shows chances of rain

## 2 Methodology

### 2.1 Notation & Background

Elementary symmetry polynomials are widely used in our paper. $\mathbb{R}[x]$ denotes the set of polynomials with coefficient in $\mathbb{R}$. $\mathbb{R}[x]_d$ are polynomials of degree no larger than $d$. We use $\odot$ for polynomial multiplication. For two polynomials $a$ and $b$, $\lfloor \frac{a}{b} \rfloor$ is the quotient of the polynomial division $a/b$.

For two vectors $x, y \in \mathbb{R}^d$, $\langle x, y \rangle$ is the inner product of $x$ and $y$. We abuse the notation, so when a polynomial appears in the inner product, we take it to mean the coefficient vector of the polynomial. Namely, if $p, q$ are both polynomials of the same degree, then $\langle p, q \rangle$ is the inner product of their coefficient vectors. We use $\cdot$ for matrix multiplication.

We refer to $x \in X \subset \mathbb{R}^m$ as an instance and $f : X \to \mathbb{R}$ as the fitted tree model in a supervised learning task. Here, $m$ denotes the number of all features, $M$ is the set of all features, and $|.|$ is the cardinality operation, namely $|M| = m$. We denote $x[i]$ as the value of feature $i$ of instance $x$.

We have to start with some common terminologies because our algorithm is closely involved with trees. A rooted tree $T = (V, E)$ is a directed tree where each edge is oriented away from the root $r \in V$. For each node $v$, $P_v$ is the root to $v$ path, i.e. the set of edges from the root to $v$. $L(v)$ is the set of leaves reachable from $v$. $L(T) = L(r)$ is the set of leaves of $T$. $T$ is a full binary tree if every non-leaf node has two children. If an edge $e$ goes from $u$ to $v$, then $u$ and $v$ are the tail and head of $e$, respectively. We write $h(e)$ for the head of $e$.

A tree is weighted, if there is an edge weight $w_e$ for each edge $e$. It is a labeled tree if each edge also has a label $\ell_e$. For a labeled tree $T$, let $E_i$ be all the edges of the tree with label $i$. Similarly, define $P_{i,v} = P_v \cap E_i$, the set of edges in the root to $v$ path that has label $i$. The last edge of any subset of a path is the edge furthest away from the root.

For our purpose, a decision tree is a weighted labeled rooted full binary tree. There is a corresponding decision tree for the fitted tree model $T_f$. The internal nodes of the tree are called the decision nodes, and the leaves are called the end nodes. Every decision node has a label of feature $i$, and every end node contains a prediction $v$. The label of each edge is the feature of the head node of the edge. We will call the label on the edge of the feature. Every edge $e$ contains weight $w_e$ that is the conditional probability based on associated splitting criteria during training.

When predicting a given instance, $x$, decision tree model $f$ sends the instance to one of its leaves according to splitting criteria. We draw an example decision tree in Fig 1. Each leaf node is labeled with id and prediction value. Every decision node is labeled with the feature. We also associate each edge with conditional probability $w$ and splitting criteria of features in the parenting node.

To represent the marginal effect, we use $f_S(x) : X \rightarrow \mathbb{R}, S \subseteq M$ to denote the prediction of instance $x$ of the fitted tree model using only the features in *active set* $S$, and treat the rest of features of instance $x$ as missing. Using this representation, the default prediction $f(x)$ is a shorthand for $f_M(x)$.

The *Shapley value* of a decision tree model $f$ is the function $\phi(f, i) : X \rightarrow \mathbb{R}$,

$$\phi(f, i)(x) = \frac{1}{|M|} \sum_{S \subseteq M \setminus \{i\}} \frac{1}{\binom{|M|-1}{|S|}} f_{S \cup \{i\}}(x) - f_S(x). \tag{1}$$

The Shapley value $\phi(f, i)(x)$ quantifies the marginal contribution of feature $i$ in the tree model $f$ when predicting instance $x$. The problem of computing the Shapley values is the following algorithmic problem.

---

**The Tree Shapley Value Problem**
**Input:** A decision tree $T_f$ for function $f : X \rightarrow \mathbb{R}$ over $m$ features, and $x \in X$.
**Output:** The vector $(\phi(f, 1)(x), \ldots, \phi(f, m)(x))$.

---

Meanwhile, decision nodes cannot split instances with missing feature values. A common convention is to use conditional expectation. When a decision node encounters a missing value, it redirects the instance to both children and returns the weighted sum of both children's predictions. The weights differ between decision nodes and are empirical instance proportions during training: $w_l, w_r$. Here $w_l + w_r = 1$ and $0 < w_l < 1$. A similar approach is also used in both Treeshap[5], and C4.5 [7] to deal with missing values. Any instance would result in a single leaf when there is no missing feature. In contrast, an instance might reach multiple different leaves with a missing feature.

Here, we use an example instance $x =$**(temperature: 20, cloudy: no, wind speed: 6)** with tree $f$ in Fig 1 to show the full process of Shapley value computing. By following the decision nodes of $T_f$, the prediction $f(x)$ is leaf $C$: 0.4 chance of raining. Now we compute the importance/Shapley value of feature **(temperature: 18)** among $x$ for getting prediction of 0.4 chance of raining.

The importance/Shapley value of feature **(temperature: 18)** is

$$\phi(f, \textbf{temperature})(x) = \frac{1}{3} \Big( \frac{1}{\binom{2}{2}} (f_{\{\textbf{temperature, cloudy, wind speed}\}}(x) - f_{\{\textbf{cloudy, wind speed}\}}(x))$$

$$+ \frac{1}{\binom{2}{1}} (f_{\{\textbf{temperature, wind speed}\}}(x) - f_{\{\textbf{wind speed}\}}(x) + f_{\{\textbf{temperature, cloudy}\}}(x) - f_{\{\textbf{cloudy}\}}(x))$$

$$+ \frac{1}{\binom{2}{0}} (f_{\{\textbf{temperature}\}}(x) - f_\emptyset(x)))$$

To elaborate more, a term $f_{\{\textbf{cloudy, wind speed}\}}(x)$ with $x =$**(temperature: 20, cloudy: no, wind speed: 6)** is equivalent to $f$(**cloudy: no, wind speed: 6**). When traversal first decision node: temperature, value to current feature is considered as unspecified. We sum over children leaves with empirical weights and get $0.5 \cdot D + 0.5 \cdot C$ as prediction.

On the other hand, decision tree $f$ can be linearized into decision rules [8]. A decision rule can be seen as a decision tree with only a single path. A decision rule $R^v : X \rightarrow \mathbb{R}$ for a leaf $v$ can be constructed via starting from root node, following all the conditions along the path to the leaf $v$. We use $F(R)$ to represent the set of all features specified in decision rule $R$, namely, $F(R) = \{i | P_{i,v} \neq \emptyset\}$.

The linearization of the decision tree $f$ to decision rules is the relation $f(x) = \sum_{v \in L(T_f)} R^v(x)$. Example tree in Fig 1 can be linearized into 4 rules:

1. $R^A$: **if** (temperature $> 19$) and (is cloudy) **then** predict 0.7 chance of rain **else** predict 0 chance of rain

2. $R^B$: **if** (temperature $> 19$) and (is not cloudy) and (wind speed $> 8$) **then** predict 0.6 chance of rain **else** predict 0 chance of rain

3. $R^C$: **if** (temperature $> 19$) and (is not cloudy) and (wind speed $\leq 8$) **then** predict 0.4 chance of rain **else** predict 0 chance of rain

4. $R^D$: **if** (temperature $\leq 19$) **then** predict 0.5 chance of rain **else** predict 0 chance of rain

For a decision rule $R$, we also introduce *prediction with active set $S$, $R_S : X \to \mathbb{R}$*. When features are missing, leaf value further weighted by their conditional probability is provided as the prediction. Here we introduce the definition recursively. First, we define the prediction of rule $R$ associated with leaf value $\mathcal{V}$ with empty input:

$$R_\emptyset^v = R_\emptyset^v(x) = \mathcal{V} \prod_{e \in P_v} w_e \tag{2}$$

Where $w_e$ is the conditional probability/proportion of instances, when splitting by decision node at the source of edge $e$, the proportion of instances belong to the current edge. $\mathcal{V}$ is the prediction of the leaf node $v$ that defines that decision rule.

We say $x \in \pi_i(R)$, if $x[i]$ is satisfied by every splitting criteria concerning feature $i$ in decision rule $R$. For a given instance $x$ and leaf $v$, we use a new variable $q_{i,v}(x)$ to denote the marginal prediction of $R^v$ when adding feature $i$ to active set $S$.

$$q_{i,v}(x) := \begin{cases} \prod_{e \in P_{i,v}} \frac{1}{w_e} & x \in \pi_i(R^v) \\ 0 & x \notin \pi_i(R^v) \end{cases} \tag{3}$$

The empty product equals 1, hence if $P_{i,v} = \emptyset$, $q_{i,v}(x) = 1$. We omit the super/subscript $v$ if there is no ambiguity on the leaf node. So, with $i \notin S$, we can write:

$$R_{\{i\} \cup S}(x) = q_i(x) R_S(x) \tag{4}$$

Since $\emptyset$ is a subset of any set $S$, we can get $R_S(x)$ via products of weights:

$$R_S(x) = R_\emptyset \prod_{j \in S} q_j(x) \tag{5}$$

With $R_S$, $f_S$ can also be linearized into the sum of rule predictions:

$$f_S(x) = \sum_{v \in L(T_f)} R_S^v(x). \tag{6}$$

## 2.2 Some special functions and their properties

**Definition 2.1.** *Define the reciprocal binomial polynomial to be $B_d(x) = \sum_{i=0}^d \binom{d}{i}^{-1} x^i$.*

**Definition 2.2.** *The function $\psi_d : \mathbb{R}[x]_d \to \mathbb{R}$ is defined as*

$$\psi_d(A) := \frac{\langle A, B_d \rangle}{d+1}. \tag{7}$$

*We write $\psi(p) = \psi_d(p)$ where $d$ is the degree of $p$.*

The function $\psi_d$ has two nice properties: additive for same degree polynomial and "scale" invariant when multiplying binomial coefficient.

**Proposition 2.1.** *Let $p, q \in \mathbb{R}[x]_d$, and $k \in \mathbb{N}$.*

- *Additivity: $\psi_d(p) + \psi_d(q) = \psi_d(p + q)$.*

- *Scale Invariant: $\psi(p \odot (1 + y)^k) = \psi(p)$.*

## 2.3 Summary polynomials and their relation to Shapley value

Consider we have a function $f$ represented by a decision tree $T_f$. We want to explain a particular sample $x$, therefore in the later sections, we abuse the notation and let $g$ to mean $g(x)$ whenever $g : X \to \mathbb{R}$, e.g $q_{i,v} = q_{i,v}(x)$. In order to not confuse the readers, the polynomials we are constructing always have the formal variable $y$.

Since tree prediction can be linearized into decision rules, and the Shapley value also has Linearity property, we decompose the Shapley value of a tree as the sum of the Shapley value of decision rules.

$$\phi(f, i) = \sum_{v \in L(T_f)} \phi(R^v, i) \tag{8}$$

Now, for each decision rule, we define a summary polynomial.

**Definition 2.3.** *For a decision tree $T_f$ and an instance $x$. For a decision rule associated with leaf $v$ in $T_f$, the* summary polynomial $G_v$ *is defined as*

$$G_v(y) = R_\emptyset^v \prod_{j \in F(R^v)} (q_{j,v} + y) \tag{9}$$

Next, we study the relationship between the summary polynomial and the Shapley value of corresponding decision rule.

**Lemma 2.2.** *Let $v$ be a leaf in $T_f$, then*

$$\phi(R^v, i) = (q_{i,v} - 1)\psi\left(\frac{G_v}{q_{i,v} + y}\right).$$

*Proof.* Since everything involved in the proof is related to the leaf $v$, we drop $v$ from the super/subscripts for simplicity. First, we simplify the Shapley value of rule $R$ into:

$$\phi(R, i) = \frac{1}{m} \sum_{S \subset M \setminus \{i\}} \frac{1}{\binom{m-1}{|S|}} R_{S \cup \{i\}} - R_S = \frac{R_\emptyset(q_i - 1)}{m} \sum_{S \subset M \setminus \{i\}} \frac{1}{\binom{m-1}{|S|}} \prod_{j \in S} q_j \tag{10}$$

When feature $i$ does not appear in $R$, $q_i - 1$ returns 0, thus Shapley value on feature $i$ from rule $R$ is 0. Let $|F(R)| = d$, the number of features in $R$. The Shapley value of $R$ further reduces to:

$$\phi(R, i) = \frac{R_\emptyset(q_i - 1)}{d} \sum_{k=0}^{d-1} \frac{1}{\binom{d-1}{k}} \sum_{\substack{S \subset F(R) \setminus \{i\} \\ |S|=k}} \prod_{j \in S} q_j \tag{11}$$

We observe that $R_\emptyset \sum_{S \subset F(R) \setminus \{i\}}^{|S|=k} \prod_{j \in S} q_j$ is precisely the coefficient of $y^k$ in $\frac{G}{q_i + y}$.

We obtain the weighted sum of all subsets' decision rule prediction using the inner product:

$$R_\emptyset \sum_{S \subset F(R) \setminus \{i\}} 1 / \binom{d-1}{|S|} \sum_{\substack{S \subset F(R) \setminus \{i\} \\ |S|=k}} \prod_{j \in S} q_j = \langle \frac{G}{q_i + y}, B_{d-1} \rangle \tag{12}$$

Shapley value for $R$ has a compact form as shown in Eq.13.

$$\begin{aligned}
\phi(R, i) &= \frac{R_\emptyset(q_i - 1)}{d} \sum_{S \subset F(R) \setminus \{i\}} \frac{1}{\binom{d-1}{|S|}} \prod_{j \in S} q_j \\
&= \frac{(q_i - 1)}{d} \langle \frac{G}{q_i + y}, B_{d-1} \rangle \\
&= (q_i - 1)\psi\left(\frac{G}{q_i + y}\right)
\end{aligned} \tag{13}$$

$\square$

### 2.4 Computations

Even though we have simplified the Shapley value of a decision rule in a compact form using polynomials, it is still not friendly in computational complexity. In particular, the values $q_{i,v}$ are flat aggregated statistics and do not necessarily share terms in-between different rules. This makes it difficult to share intermediate results across different rules. To benefit from the fact that decision rules overlap, we develop an edge-based polynomial representation.

For every edge $e$ with feature $i$, we use $e^\uparrow$ to denote its closest ancestor edge that shares the same feature. In cases such edge does not exist, $e^\uparrow = \perp$. We also use $x \in \pi_u$ to represent the $x[i]$ is satisfied by all splitting criteria on feature $i$ associated with all edges in $P_{i,u}$.

$$p_e := \begin{cases} \prod_{e' \in P_{i,h(e)}} \frac{1}{w_{e'}} & x \in \pi_{h(e)} \\ 0 & x \notin \pi_{h(e)} \end{cases} \tag{14}$$

We also define additionally that $p_\perp = 1$.

If $e$ is the last edge in $P_{i,v}$ then $p_e = q_{i,v}$. This is the key to avoid $q_{i,v}$ completely, and instead switch to $p_e$. Our analysis will make sure that any $p_e$ that does not correspond to $q_{i,v}$ for any $v$ and $i$ gets cancelled out.

We show a relation between the Shapley value of a decision rule and the newly defined $p_e$'s.

Consider an operation $\oplus_{d_1,d_2} : \mathbb{R}[x]_{d_1} \times \mathbb{R}[x]_{d_2} \to \mathbb{R}[x]_{\max(d_1,d_2)}$. The subscript is omitted when $d_1, d_2$ is implicit.

$$G^1 \oplus G^2 := G^1 + G^2 \odot (1+y)^{d_1-d_2}, \tag{15}$$

We extend the summary polynomial to all nodes in the tree. Let $G_u = \bigoplus_{v \in L(u)} G^v$. Denote $d_e$ as the degree of $G_u$, where $h(e) = u$.

**Proposition 2.3.** *Let $v$ be a leaf in $T_f$, and $d_v$ be the degree of $G_v$ then*

$$\phi(R^v, i) = \sum_{e \in P_{i,v}} (p_e - 1)\psi\left(\left\lfloor \frac{G_v \odot (y+1)^{d_e-d_v}}{y+p_e} \right\rfloor\right) - (p_{e^\uparrow} - 1)\psi\left(\left\lfloor \frac{G_v \odot (y+1)^{d_{e^\uparrow}-d_v}}{y+p_{e^\uparrow}} \right\rfloor\right).$$

*Proof.* Let $e^*$ be the last edge of $P_{i,v}$. We note a few facts. $p_{e^*} = q_{i,v}$, $y + q_{i,v}$ divides $G_v$, and the sum is a telescoping sum. Put them together.

$$\sum_{e \in P_{i,v}} (p_e - 1)\psi\left(\left\lfloor \frac{G_v \odot (y+1)^{d_e-d_v}}{y+p_e} \right\rfloor\right) - (p_{e^\uparrow} - 1)\psi\left(\left\lfloor \frac{G_v \odot (y+1)^{d_{e^\uparrow}-d_v}}{y+p_{e^\uparrow}} \right\rfloor\right)$$

$$= (p_{e^*} - 1)\psi\left(\left\lfloor \frac{G_v \odot (y+1)^{d_{e^*}-d_v}}{y+p_{e^*}} \right\rfloor\right)$$

$$= (q_{i,v} - 1)\psi\left(\left\lfloor \frac{G_v \odot (y+1)^{d_{e^*}-d_v}}{y+q_{i,v}} \right\rfloor\right)$$

$$= (q_{i,v} - 1)\psi\left(\frac{G_v}{y+q_{i,v}} \odot (y+1)^{d_{e^*}-d_v}\right)$$

$$= (q_{i,v} - 1)\psi\left(\frac{G_v}{y+q_{i,v}}\right)$$

$$= \phi(R^v, i)$$

$\square$

The following theorem establishes the relation between Shapley values, the summary polynomials at each node, and $p_e$ for each edge $e$.

**Theorem 2.4** (Main)**.**

$$\phi(f, i) = \sum_{e \in E_i} (p_e - 1)\psi\left(\left\lfloor \frac{G_{h(e)}}{y+p_e} \right\rfloor\right) - (p_{e^\uparrow} - 1)\psi\left(\left\lfloor \frac{G_{h(e)} \odot (y+1)^{d_{e^\uparrow}-d_e}}{y+p_{e^\uparrow}} \right\rfloor\right)$$

*Proof.* Based on linearity of Shapley Value, $\phi(f, i) = \sum_{v \in L(T_f)} \phi(R^v, i)$.

For each rule $R^v$, we can scale their summary polynomial $G_v$ to the degree of $G_{h(e)}$. Based on Proposition 2.3,

$$\phi(f, i) = \sum_{v \in L(T_f)} \sum_{e \in P_{i,v}} (p_e - 1)\psi\left(\left\lfloor \frac{G_v \odot (y+1)^{d_e - d_v}}{y + p_e} \right\rfloor\right) - (p_{e\uparrow} - 1)\psi\left(\left\lfloor \frac{G_v \odot (y+1)^{(d_e - d_v) + (d_{e\uparrow} - d_e)}}{y + p_{e\uparrow}} \right\rfloor\right)$$

Observe that for any $(e, v)$ pair, we have $e \in E_i$ and $v \in L(h(e))$ if and only if $v \in L(T_f)$ and $e \in P_{i,v}$. Hence, can order the summation by summing through the edges.

$$\phi(f, i) = \sum_{e \in E_i} \sum_{v \in L(h(e))} (p_e - 1)\psi\left(\left\lfloor \frac{G_v \odot (y+1)^{d_e - d_v}}{y + p_e} \right\rfloor\right) - (p_{e\uparrow} - 1)\psi\left(\left\lfloor \frac{G_v \odot (y+1)^{(d_e - d_v) + (d_{e\uparrow} - d_e)}}{y + p_{e\uparrow}} \right\rfloor\right)$$

Observe that at each edge $e$, all summary polynomial $G$ is scaled to the same degree $d_e$. According to Proposition 2.1, we can add the summary polynomials before evaluate using $\psi(.)$. Now, focus on the first part of the sum.

$$\sum_{e \in E_i} \sum_{v \in L(h(e))} ((p_e - 1)\psi(\lfloor \frac{G_v \odot (y+1)^{d_e - d_v}}{y + p_e} \rfloor) = \sum_{e \in E_i} (p_e - 1)\psi\left(\sum_{v \in L(h(e))} \left\lfloor \frac{G_v \odot (y+1)^{d_e - d_v}}{y + p_e} \right\rfloor\right)$$

$$= \sum_{e \in E_i} (p_e - 1)\psi(\lfloor \frac{\sum_{v \in L(h(e))} G_v \odot (y+1)^{d_e - d_v}}{y + p_e} \rfloor)$$

$$= \sum_{e \in E_i} (p_e - 1)\psi(\lfloor \frac{\bigoplus_{v \in L(h(e))} G_v}{y + p_e} \rfloor)$$

$$= \sum_{e \in E_i} (p_e - 1)\psi\left(\left\lfloor \frac{G_{h(e)}}{y + p_e} \right\rfloor\right)$$

Using the exact same proof, we can also obtain

$$\sum_{e \in E_i} \sum_{v \in L(h(e))} (p_{e\uparrow} - 1)\psi\left(\left\lfloor \frac{G_v \odot (y+1)^{(d_e - d_v) + (d_{e\uparrow} - d_e)}}{y + p_{e\uparrow}} \right\rfloor\right) = \sum_{e \in E_i} (p_{e\uparrow} - 1)\psi\left(\left\lfloor \frac{G_{h(e)} \odot (y+1)^{d_{e\uparrow} - d_e}}{y + p_{e\uparrow}} \right\rfloor\right)$$

$\square$

## 2.5   Linear TreeSHAP and complexity analysis

By Theorem 2.4, we can obtain an algorithm in two phases. Efficiently compute the summary polynomial on each node(Algorithm 2) and then evaluates for $\phi(f, i)$ directly(Algorithm 3). Both parts of the algorithm are straightforward, basically computing directly through definition and tree traversal. The final values of $S[i]$ is the desired value $\phi(f, i)(x)$ after running Algorithm 4.

To analyze the running time, one can see each node is visited a constant number of times. The operations are polynomial addition, multiplication, division, inner product, or constant-time operations. In general, all those polynomial operations takes $O(D \log D)$ time for degree $D$ polynomial [1]. This shows the total running time is $O(LD \log D)$.

However, we never need the coefficients of the polynomials. So we can improve the running time by storing the summary polynomials in a better-suited form, the multipoint interpolation form. Namely, we evaluate the polynomials $G$ on a set of predefined unique points $Y = (y_0, y_1, y_2, \cdots, y_D) \in \mathbb{R}^{D+1}$, and store $G(Y) = (G(y_0), \ldots, G(y_D))$ instead. In this form, addition, product and division takes $O(D)$ time [2]. The evaluation function $\psi(G)$ also takes $O(D)$ time but needs more explanation.

Denote $\mathcal{V}(Y) \subset \mathbb{R}^{D+1 \times D+1}$ as the Vandermonde matrix of $Y$, where $v_{i,j} \in \mathcal{V}(Y) = y_i^j$ is the $j$th power of $y_i$.

```
COMPUTESUMMARYPOLYNOMIALS(x, v, C):
    if node v is leaf:
        G[v] ← C · R_∅^v
    else:
        if v is not the root:
            e ← edge with v as head
            C ← C ⊙ (y + p_e(x))
            if e^↑ ≠ ⊥:
                C ← C / (y + p_{e^↑}(x))
        l, r ← children of v
        COMPUTESUMMARYPOLYNOMIALS(x, l, C)
        COMPUTESUMMARYPOLYNOMIALS(x, r, C)
        G[v] ← G[l] ⊕ G[r]
    return G[v]
```

**Algorithm 2:** Obtain the summary polynomial for each node.

```
AGGREGATESHAPLEY(x, v, G):
    if v has children:
        l, r ← children of v
        AGGREGATESHAPLEY(x, l, G)
        AGGREGATESHAPLEY(x, r, G)
    if v is not the root:
        e edge with v as head
        i ← feature of edge e
        S[i] ← S[i] + (p_e(x) − 1)ψ(⌊ G[v] / (y + p_e(x)) ⌋)
        S[i] ← S[i] − (p_{e^↑}(x) − 1)ψ(⌊ G[v] ⊙ (y + 1)^{d_{e^↑} − d_e} / (y + p_{e^↑}(x)) ⌋)
```

**Algorithm 3:** Obtain the Shapley value vector.

**Lemma 2.5.** *Let $p, q \in R[x]_d$, and its coefficients $A$ and $B$, respectively, then we have*

$$\langle p, q \rangle = \langle A, B \rangle = \langle p(Y), \mathcal{V}(Y)^{-1}B \rangle.$$

*Proof.* Polynomial evaluation can be consider as inner product of coefficient and Vandermonde matrix of input $Y$. Namely $p(Y) = A \cdot \mathcal{V}(Y)$. Therefore $\langle p(Y), \mathcal{V}(Y)^{-1}B \rangle = \langle A \cdot \mathcal{V}(Y), \mathcal{V}(Y)^{-1}B \rangle = \langle A, \mathcal{V}(Y) \cdot \mathcal{V}(Y)^{-1}B \rangle = \langle A, B \rangle$ completes the proof. □

In order to compute the inner products $\langle G, B_d \rangle$ in $O(D)$ time, we have to precompute $N_d = \mathcal{V}(Y)^{-1}C_d$, where $C_d$ is the coefficient of $B_d$, for all $0 \le d \le D$. This can be done simply in $O(D)$ time for each $d$, so a total of $O(D^2)$ time.

By storing the polynomial in interpolation form, all our polynomial operations on each node take $O(D)$ time. Therefore the total running time is $O(LD)$.

Other than the summary polynomials, the algorithm uses constant space to store information on nodes and edges. Each summary polynomial takes $O(D)$ space to store. Therefore the algorithm takes $O(LD)$ space. Nevertheless, we can save space by realizing the algorithms only need a single top-down and a single bottom-up step. By joining two steps into one, the algorithm consumes the summary polynomials on the spot. Hence the total space usage will be bounded by $O(D)$ times the stack size, bounded by $D$, the depth of the tree. The final total space usage is improved to $O(D^2)$.

**Remark** Even though $Y$ can be arbitrarily chosen based on the maximum depth of the tree $D$, it is shown that Chebyshev points are near-optimal in numerical stability [11]. In our Linear TreeShap implementation, we used the Chebyshev points of the second kind.

```
LINEARTREESHAP(x, T_f):
    G ← an array indexed by the nodes
    COMPUTESUMMARYPOLYNOMIALS(x, root(T_f), 1)
    S ← an array indexed by the features
    AGGREGATESHAPLEY(x, root(T_f), G)
    return S
```

**Algorithm 4:** The entire LINEARTREESHAP algorithm.

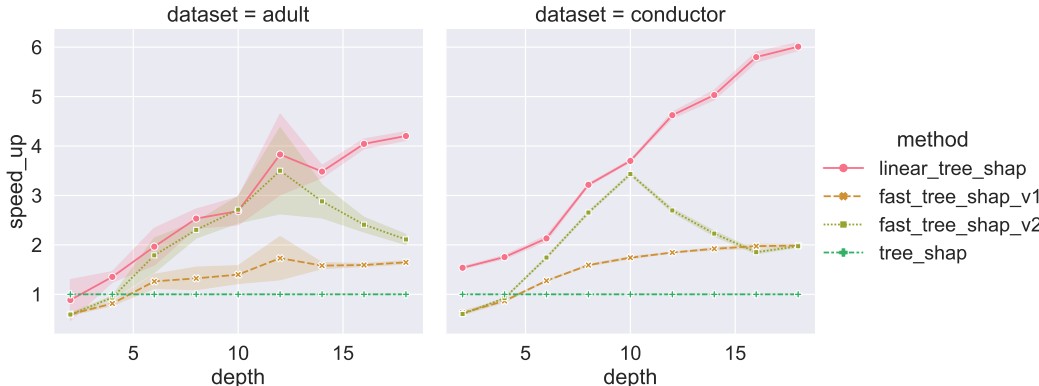

**Figure 5:** Speed up comparison

| Datasets | # Instances | # Attributes | Task | Classes |
|---|---|---|---|---|
| adult [4] | 48,842 | 64 | Classification | 2 |
| conductor [3] | 21,263 | 81 | Regression | - |

**Table 2:** Datasets

## 3   Experiments

We run an experiment on both regression dataset adult and classification dataset conductor(summary in Table 2) to compare both our method and two popular algorithms, TreeShap and Fast TreeShap. We explain Trees with depths ranging from 2 to 18. And to align the performance across different depths of trees, we plot the ratio between the time of Tree Shap and the time of all methods in Figure 5. We run every algorithm on the same test set 5 times to get both average speeds up and the error bar. And for fair comparison purposes, all methods are limited to using a single core.

Linear TreeShap is the fastest among all setups. And due to heavy memory usage, Fast TreeShap V2 falls back to V1 when tree depth reaches 18 for the dataset conductor. Since the degree of the polynomial is bounded both by the depth of the tree also the number of unique features per decision rule, with deeper depth, dataset Conductor has much more speed up gains thanks to a higher number of features. We can conclude that the Linear TreeShap is more efficient than all state-of-the-art Shapley value computing methods in both theory and practice.

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
