# OpenReview forum: "Linear tree shap"
_NeurIPS.cc/2022/Conference — NeurIPS 2022 Accept_

### Official Review · Reviewer_sVJC · 2022-07-10

**Rating:** 7
**Confidence:** 3
**Soundness:** 4 excellent
**Presentation:** 4 excellent
**Contribution:** 4 excellent

**Summary:**

The authors present a new exact method to compute Shapley values for decision trees that can be computed in O(SLD) where S is the number of samples, L is the number of leaves and D is the maximum depth of the tree. This compares to previous exact methods operating in O(SLD^2). The authors provide both the mathematical proof for their method as well as empirical comparisons to previous algorithm to compute Shapley values.

**Questions:**

I have no further questions or suggestions for this work.

**Limitations:**

There are no obvious negative societal effects of this work.

**Strengths And Weaknesses:**

The paper is well-written and motivated and provides a clear improvement over previously available methods to compute Shapley values for decision trees. Decision trees are an important non-linear prediction tool and Shapley values integral in analyzing their fit and selecting variables. Hence, I believe this paper makes a significant and original contribution to the field.

---

### Official Review · Reviewer_yep3 · 2022-07-12

**Rating:** 7
**Confidence:** 4
**Soundness:** 4 excellent
**Presentation:** 2 fair
**Contribution:** 3 good

**Summary:**

This work considers the problem of calculating Shapley values for tree-based models. Without a clever algorithm, calculating Shapley values has exponential running time, but TreeSHAP introduced a procedure for making the running time polynomial. The approach presented here, Linear TreeSHAP, aims to design a procedure that is even more efficient (without compromising on memory).

The approach is quite technical, but if the authors' results are correct, then Linear TreeSHAP achieves exact Shapley value calculation with better time complexity in theory (by a factor of $D$, the maximum tree depth) and a noticeable speedup in practice (which grows larger with deeper trees). The tools used here are interesting, and to my knowledge novel in this subfield.

**Questions:**

Several questions about the presentation and experiments are included above.

**Limitations:**

There are no negative societal impacts for this work.

The authors give a detailed description of their algorithm's run-time and memory complexity, describing the most expensive computations. The only additional material that might highlight limitations would be an expansion of the experiments section.

**Strengths And Weaknesses:**

### Strengths

- The authors present a new procedure for calculating Shapley values for tree-based models. They provide results showing that their algorithm is exact (it involves no statistical estimation) and efficient (the time complexity analysis yields a linear rather than quadratic dependence on the tree depth). The speedup is reflected in practice
- The authors make use of interesting technical tools (summary polynomials) to design their approach

### Weaknesses

The presentation was difficult to follow. This is perhaps inevitable for a technical approach like this, but there were a couple places where it could be improved:
- $m$ and $N$ are both used to denote the number of features
- The process of linearizing a tree could be described specifically around lines 91-94, as it's quite simple - i.e., there's a single decision node with a split based on all edges in $P_v$, with the prediction being either the original node prediction or zero
- "Marginal prediction" is a confusing name for $q_{i, v}(x)$. That name sounds like "marginal contribution" from the game theory context, but to get the marginal contribution we actually need $R_S(x)(q_{i, v}(x) - 1)$, so the multiplier itself is not the marginal contribution
- In equation 6, this isn't a completely obvious result so it may be helpful to include something like "according to this definition of $R_S$, we have the following equivalence with the previously defined $f_S$"
- I attempted to verify the math up until a certain point, and I found equation 11 somewhat difficult to derive from equation 10. I don't think this result is described in the main text or appendix, could it be added?

Some other notes on presenting the approach:
- What's the motivation for the name "linear TreeSHAP," is it the reduced dependence on $D$ in the complexity analysis?
- In the introduction, would it be possible to give more intuition for how the approach works instead of "we solve the exact Shapley value computing problem based on polynomial arithmetic" ? I'm not sure how I would refine this, but it's a pretty vague description of what's presented in later sections
- Could the authors add some discussion about how their approach differs, in its derivation and the techniques involved (rather than just the run-time), from TreeSHAP and FastTreeSHAP? For readers who aren't familiar with the details of those algorithms, even a high-level discussion would be helpful

About the experiments:
- The results are quite positive. Would it be possible to show how the different algorithms improve when parallelizing across multiple cores?
- Would it be possible to add a couple more datasets, for example those shown in the FastTreeSHAP work?
- Because the math is difficult to verify, it would be very helpful to show experimentally that the three algorithms (TreeSHAP, FastTreeSHAP, Linear TreeSHAP) yield identical results

Nits:
- On line 5, "provides a linear weighting" is an unusual description for SHAP. In what sense is it linear? The attributions are perhaps *additive*, in that they sum to the prediction (minus the base rate prediction) and are derived from a weighted least squares problem
- On lines 22-23: these criticisms of prior work don't make sense. How are either GPUTreeSHAP or FastTreeSHAP lacking mathematical foundation? GPUTreeSHAP is based on the original TreeSHAP algorithm, and FastTreeSHAP is mathematically justified as well. How are they "empirical" approaches? That sounds like it means heuristics that aren't necessarily correct or well understood, which isn't true here. And how are they harder to understand than Linear TreeSHAP?

---

> ### Author Response · Authors · 2022-07-28
> **Thanks for the minor comments and answered all the questions.**
>
> Thanks for your detailed review. We will incorporate the minor comments in the final version.
>
> And here are our answers to your questions
>
> - Derive equation 11  from equation 10:
>
> We combined two steps into one to save space. The first step is to replace M with F(R). Since m is the number of features in the dataset and d = |F(R)| is the number of features in the specific decision rule. We can ignore features not specified in the rule.
> And the second step is to partition all the subsets of F(R)  based on size ranging from 0 to d-1.
>
> - Motivation for the name "linear TreeSHAP,"
>
> Yes, the motivation is because of reduced dependence on D in the complexity analysis.
>
> - More intuition for how the approach works:
>
> Shapley value of decision tree is defined combinatorically. We utilize the connection between polynomials and combinatorics and caching-friendly tree structure.
>
> - Some discussion about how our approach differs in its derivation and the techniques from TreeSHAP and FastTreeSHAP involved:
>
> In the framework of Linear Treeshap,  recall Linear Treeshap has a top-down and a bottom-up tree traversal; Treeshap is essentially only doing top-down tree traversal and then enumerating all the leaves to root paths independently. As for Fast Treeshap, it is doing almost the same as Treeshap, except for enumerating all the possible paths in the beginning.
>
> - The results are quite positive. Would it be possible to show how the different algorithms improve when parallelizing across multiple cores?
>
> Typically, the parallelization is on the instance level. With multiple cores/machines, we should see a linear speedup proportional to the number of cores for all the methods compared. Even for the fast treeshap v2, the pre-computing time is omitted from the comparison.
>
> - Would it be possible to add a couple more datasets, for example those shown in the FastTreeSHAP work?
>
> Yeah, of course, we can, and there shouldn't be any different. Due to limited space, we didn't add more but would probably extend this paper in some journal format and include more datasets. Also, there are aspects of numerical stability of different methods we didn't have space to elaborate on.
>
> - Because the math is difficult to verify, it would be very helpful to show experimentally that the three algorithms (TreeSHAP, FastTreeSHAP, Linear TreeSHAP) yield identical results.
>
> Yes, we did verify that all of them match up to 5-digit precision and the code and results are provided in supplementary materials.

---

> > ### Comment · Reviewer_yep3 · 2022-08-07
> > **Thanks for response**
> >
> > Thanks to the authors for their response. A couple additional thoughts:
> >
> > **About equation 11.** I get that the two steps are replacing $M$ with $F(R)$ and partitioning subsets based on their size. The part that would be nice to reproduce is how you derive the new weights for each $S \subseteq F(R) \setminus i$ in the updated summation: they change from $\frac{1}{m} * \binom{m - 1}{|S|}^{-1}$ in eq. 10 to $\frac{1}{d} * \binom{d - 1}{|S|}$ in eq. 11. It seems that this should be done by counting the ways to take $S \subseteq F(R) \setminus i$ and create subsets $W = S \cup T$ for $T \subseteq M \setminus (F(R) \cup i)$ with different sizes $|W| = h$, and then summing the weights across $h = |S|, |S| + 1, ..., m - (d - |S|) - 1$. This is non-trivial and readers should not be forced to derive it from scratch, please include it in the appendix.
> >
> > **About the name "Linear TreeSHAP."** Like reviewer 1DKp, I'm not fond of the name as it does not reflect the new ideas in the algorithm or provide an accurate/complete description of the computational complexity. I don't have a specific recommendation or request, but I would encourage the authors to rethink the name.
> >
> > **About additional datasets.** This seems like a straightforward addition to the supplement that you can reference in the experiments section.
> >
> > **About the consistency between TreeSHAP variants.** This also seems like a straightforward addition to the supplement, and this would be more helpful than including it in the supplementary code. It probably doesn't need to be in the main text for this version or any future journal version of the work, but you could add it there and reference it in the experiments section.
> >
> > **Other presentation improvements.** Please see the "weaknesses" section of my review for a couple simple writing changes that would make the paper easier to follow.
> >
> > My overall view of the paper remains positive, so I'm keeping my score as is.

---

### Official Review · Reviewer_nHUS · 2022-07-13

**Rating:** 7
**Confidence:** 3
**Soundness:** 3 good
**Presentation:** 3 good
**Contribution:** 3 good

**Summary:**

- The authors proposed a novel algorithm that improves the computational complexity of TreeShap with the same amount of memory.
- TreeShap has been a popular algorithm for understanding ensemble of decision trees. The proposed algorithm will have some applications for large-scale data.


**Questions:**

- Will the authors release the code and add it to the official TreeShap github?

**Ethics Review Area:**

["I don’t know"]

**Limitations:**

The area is relatively narrow.

**Strengths And Weaknesses:**

Strengths:
- The correctness of the algorithm is rigorously verified.
- The improvement in computation is significant.
Weaknesses:
- The algorithm is a specific refinement to an algorithm in a relatively narrow area. But this is not a reason for rejection.

---

> ### Author Response · Authors · 2022-07-27
> **We will release the code on GitHub for sure**
>
> Adding to TreeShap or simply an independent GitHub repo is still not decided, but we are open to suggestions.

---

### Official Review · Reviewer_1DKp · 2022-07-17

**Rating:** 7
**Confidence:** 3
**Soundness:** 4 excellent
**Presentation:** 2 fair
**Contribution:** 3 good

**Summary:**

This paper proposes an efficient implementation of an algorithm to compute of Shapley values from decision trees. This implementation is based on polynomial arithmetic. With respect to existing algorithm it reduces the dependency of computational complexity on tree depth from D^2 to D, while it remains exact.


**Questions:**

I would suggest the authors to rewrite their introduction, to improve the experimental part and add a conclusion.


**Limitations:**

I don't see any negative societal impacts to the work (since it aims at improving model interpretabilaty). There is no discussion of the limitation of the approach. Given the nature of the contribution, a strict improvement of some existing work, it's not really a problem though.


**Strengths And Weaknesses:**

Reducing the computing times of exact TreeShap is a very nice result given the desirable theoretical properties of this metric and its popularity. So, I think the contribution of the paper is important. The derivation of the algorithm, based on polynomial arithmetic, is very clever and looks sound to me (although I didn't check every steps of the proofs and I'm also not an expert of polynomial arithmetic). In the end, the final algorithm remains rather simple to implement.

The presentation in the paper could be improved however. The introduction is very minimal and seems to have been rushed. Section 1.1 presents the computational complexity of all compared methods without any discussion. I think that Linear TreeShap should be better contrasted with respect to previous results. The paper should incorporate a more detailed presentation of related works.

In contrast, I find the description of the proposed algorithm rather clear and accurate, despite a few typos (see below). The pseudo-code is also clear. I think however that some proofs could have been moved to the supplement to gain space for more discussion in the introduction and experimental section.

The experiments are very minimal and a bit rushed again. I understand that the computational complexity of the proposed algorithm is strictly better than all other proposals, without any sacrifice in terms of exactitude and space complexity. But still, I would have been interested by an assessment of the impact of more parameters on performance, in addition to tree depth. For example, I see that Fast TreeShap v2 carries out some pre-computations to improve computing times when several explanations are computed. How does this method compare against Linear TreeShap when the number of samples to be explained increases? Reporting actual computing times would have been interesting as well.

Some details are also missing: the size of the test set used on both datasets, how the trees have been constructed (are they single trees, forests, boosted ensembles?).

Minor comments:
- I have no better name to suggest but I'm not fond of the name of the method, "linear TreeShap". Note that strictly the dependence is not linear with respect to tree complexity or depth, because it depends on the product LD.
- It should be mentioned in the paper that the proof of proposition 2.1 is in the appendix.
- Some typos:
     - Line 50: "tail and head e" => of e.
     - Line 88: "equalivant"
     - Line 94: Use $R^v$ instead of $R$?
     - Eq (2): "v" is used both to denote a leaf and its value. Maybe a different notations should be used.
     - Line 159: "last edge of in $P_{i,v}$".
     - Line 160: "any $p_e$ does not" => "any $p_e$ that does not".
     - Line 204: "to to".
     - There are many problems in the references (authors list in [2], name of proceedings is cut in [4], no journal in [7], [9] is a reference to a review of Quinlan's book, not to Quinlan's book).

---

> ### Author Response · Authors · 2022-07-28
> **Thank you for your detailed comments.**
>
> Thank you for your detailed review comments. We will adopt all the suggestions on missing experiment details and fix all the typos in the final version.
>
> For your interest, the code to reproduce the experiment and the actual running time is shared in the supplementary material.
>
> And here are the answers to the specific questions
>
> -   How does Fast TreeShap v2 compare against Linear TreeShap when the number of samples to be explained increases?
>
>
> There shouldn't be any different from the reported. As all the methods are parallelized on the instance level.  And in our reporting, the precomputing time of Fast TreeShap v2 is omitted from the comparison.

---

### Official Review · Reviewer_RR9B · 2022-07-17

**Rating:** 7
**Confidence:** 4
**Soundness:** 3 good
**Presentation:** 3 good
**Contribution:** 3 good

**Summary:**

The paper introduces "Linear TreeSHAP", a novel method to compute exact Shapley-Value for decision trees. The method leverages observations from the well-known and used TreeSHAP method, and utilizes properties of polynomials to reduce the complexity of TreeSHAP from $O(LSD^2)$ to $O(LSD)$, when $D$ is the max depth of the tree. Linear TreeSHAP is empirically compared to TreeSHAP and its more efficient variant FastTreeSHAP, and shows significant improvement in run time for deep decision trees.

**Questions:**

I would like to hear the authors' opinions on two possible limitations of the method:
1.  $\textbf{the effect of unused features}$ - consider the tree described in figure 1, and assume that there is another feature provided to the model - "season". Also, assume that we fit a tree with a low depth and that the fitted tree does not contain the season feature at all. When evaluating $E(f(x)| x_{(season, couldy)})$, the method will assume 0.5 prob. of temperature to be above 19, and 0.5 prob. to be below 19. However, it is very likely that $P(temperature < 19| season) \neq P(temperature < 19)$, so even though the model does not explicitly use season, the conditional probability-based feature elimination process that estimates  $\phi(\{season, cloudy\})$ should be effected from the knowledge that the season feature provides.

2.  $\textbf{weights balancing}$ - When training decision tree classifiers in practice, it is very common to balance the weights between the different classes. When doing so, the weights are no longer representing the conditional distribution of the samples in the train data and therefore using these weights to evaluate $f(x_s)$ might be inaccurate. I wonder if there is another interpretation of the results yielded in this scenario and if there is a way to use the training data distribution to overcome this issue.

Despite the fact that both limitations also apply to the original TreeSHAP, I believe that highlighting them in the paper will allow further research, and prevent misuse of the method.

**Limitations:**

It would be helpful if the authors will mention some limitations of their work, especially about how to interpret the output of their method. For example, it is important to mention that this method only explains the tree, and should be used carefully when trying to infer causality or conclusions on the structure of the data.

**Strengths And Weaknesses:**

Strengths:

Shapley Value based feature importance methods are widely used both in academia and in the industry. Since computing the exact Shapley Value is hard computationally-wise, introducing methods that can efficiently calculate the exact values are of high importance. This paper introduces such a solution for decision trees, which are commonly used models, especially in scenarios where explainability is required. The algorithm is based on a grounded theory that guarantees linear run time for any tree and showed both theoretically and empirically to outperform existing methods.

I also think that the authors did a good job of explaining the background and simplifying the intuition behind the algorithm.

Weaknesses:

* I think that the paper is worth publishing as is, but since its main potenital impact is to practically allow computation of Shapley Value on trees, providing more information about its limitations and assumptions might help practitioners to use it more wisely (I provide some examples in the questions section).
* In the experiments, It seems like Linear TreeSHAP showed to significantlly outperform FastTrreeSHAP only in cases where FastTrreeSHAP V2 falls back to FastTreeSHAP V1 due to memory limitations. In my opinion, it is important to show at which point Linear TreeSHAP outperforms FastTrreeSHAP V2 where there are no memory limitations.

---

> ### Author Response · Authors · 2022-07-28
> **Two cents on two possible limitations of the method**
>
> 1, Effect of unused feature
>
> You are absolutely right. Once the decision tree model is fitted, the information on unused features is lost. And the Shapley value is only defined on the fitted model instead of the fitted model and training dataset. To address this limitation, we need to redefine the Shapley value. Even though it's out of the current paper's scope, but surely worth our future study.
>
> 2, Weights balancing
>
> This is also very similar to the first question; we need to redefine the Shapley value of the decision tree. Also, an exciting future research direction. Thank you for your brilliant question/direction :)

---

### Meta-Review · Area_Chair_z2sm · 2022-08-20

**Recommendation:** Accept
**Confidence:** Certain

**Metareview:**

Shapley values are a common tool used for evaluating feature importance. In this work the authors present a way to accelerate the computation of these values when the model used is a tree or an ensemble of trees. The algorithm presented has linear computational complexity with respect to the maximal depth of the tree $D$ while previous algorithms had a computational complexity proportional to D^2 or even worse. The results are theoretically well grounded, and a small empirical study shows the merits do translate from theory to practice.
There was a consensus among reviewers that this work presents a strong scientific contribution that is relevant to NeurIPS. Some comments were made about the presentation of this work, but all agreed that the merits outweigh these limitation and therefore we recommend accepting this work to NeurIPS. Nevertheless, we encourage the authors to take a close look at the comments made by the reviewers and try to improve the presentation for the camera-ready of this work. We think that improving the presentation will improve the potential impact of this work.


**Award:**

No

---

### Decision · Program_Chairs · 2022-09-14

Accept